# Physical activity promoting teaching practices and children's physical activity within physical education lessons underpinned by motor learning theory (SAMPLE-PE)

Matteo Crotti[1,2], James Rudd[3], Simon Roberts[1], Katie Fitton Davies[1], Laura O'Callaghan[1], Till Utesch[4], Lawrence Foweather[1] *

1 Research Institute for Sport and Exercise Sciences, Liverpool John Moores University, Liverpool, United Kingdom, 2 Centre of Sport, Exercise and Life Sciences, Coventry University, Coventry, United Kingdom, 3 Norwegian School of Sport Sciences, Oslo, Norway, 4 Institute of Educational Sciences, Department of Pedagogical Assessment and Potential Development, University of Münster, Münster, Germany

* L.Foweather@ljmu.ac.uk

## Abstract

### Purpose

Movement competence is a key outcome for primary physical education (PE) curricula. As movement development in children emerges through physical activity (PA), it is important to determine the extent of PA promotion within movement competence focused teaching pedagogies. Therefore, this study aimed to assess children's moderate-to-vigorous PA (MVPA) and related teaching practices in primary PE within Linear pedagogy and Nonlinear pedagogy and to compare this to current practice within PE delivery in primary schools.

### Methods

Participants ($n$ = 162, 53% females, 5-6y) were recruited from 9 primary schools within the SAMPLE-PE cluster randomised controlled trial. Schools were randomly-allocated to one of three conditions: Linear pedagogy, Nonlinear pedagogy, or control. Nonlinear and Linear pedagogy intervention schools received a PE curriculum delivered by trained deliverers over 15 weeks, while control schools followed usual practice. Children's MVPA was measured during 3 PE lessons (44 PE lessons in total) using an ActiGraph GT9X accelerometer worn on their non-dominant wrist. Differences between conditions for children's MVPA were analysed using multilevel model analysis. Negative binomial models were used to analyse teaching practices data.

### Results

No differences were found between Linear pedagogy, Nonlinear pedagogy and the control group for children's MVPA levels during PE. Linear and Nonlinear interventions generally included higher percentages of MVPA promoting teaching practices (e.g., Motor Content) and lower MVPA reducing teaching practices (e.g., Management), compared to the control

**Data Availability Statement:** The datasets and related metadata (statistical analysis codes used to analyse the data) are publicly available within an

open access repository (link to the data and metadata: https://doi.org/10.24377/LJMU.d.00000102).

**Funding:** The authors received no specific funding for this work.

**Competing interests:** The authors have declared that no competing interests exist.

group. Teaching practices observed in Linear and Nonlinear interventions were in line with the respective pedagogical principles.

## Conclusions

Linear and Nonlinear pedagogical approaches in PE do not negatively impact MVPA compared to usual practice. Nevertheless, practitioners may need to refine these pedagogical approaches to improve MVPA alongside movement competence.

## Introduction

Physical education (PE) should provide varied, meaningful and developmentally appropriate learning experiences for children to acquire the attributes needed to lead physically active lives [1–5]. Given the well-established health benefits of participation in moderate-to-vigorous physical activity (MVPA) for children [6–9], public health arguments have been made that PE lessons should be physically active and involve teaching physical, cognitive, social and emotional skills in and through movement [10]. This health-related rationale led to the development of a goal for students to spend at least 50% of the PE lesson time engaged in MVPA [11], a guideline which has subsequently been adopted by several PE organisations across the globe [4, 12, 13]. Despite this guideline, recent research shows that students only spend between 9.5% and 42.4% of PE time engaged in MVPA [14–17]. While it is important to acknowledge that the focus on MVPA should not come at the expense of other important and meaningful PE learning outcomes [18, 19], monitoring MVPA levels during PE lessons is important to maximise physical activity (PA) opportunities during PE [19, 20].

Children's MVPA levels in PE can be affected by numerous factors, including the proportion of boys and girls in the class, lesson content (e.g., ball games, fitness, dance), and lesson location (e.g., outdoors, indoors) [14, 16, 21]. Teaching practices also play a central role in determining children's MVPA during PE lessons, through teachers' decisions on lesson content, time management (e.g., the amount of time spent explaining a task, or the amount of time before moving to a different task), and delivery (e.g., enthusiastic verbal promotion of PA engagement). Pedagogy, defined as interdependent elements of curriculum design, learning and teaching [22], is also important, with PE teachers possessing higher levels of pedagogic content knowledge (i.e., being able to deliver PE using different pedagogical approaches) and positive attitudes towards PA promotion generally being more effective in promoting PA during PE [21, 23–25]. However, there are concerns that primary PE deliverers (which often include generalist classroom teachers) do not have the required level of pedagogic content knowledge to support learning and foster student's PA [26]. Nevertheless, few studies have examined the association between different pedagogical approaches in PE and student MVPA levels. Thus, to maximise PA opportunities during PE, examining the extent to which teaching practices support students' MVPA under different pedagogical conditions is warranted.

An important feature of meaningful PE experiences and a key objective for PE curricula in young children (5-to-7-years-old) is the development of foundational movement skills needed for a lifetime of diverse PA opportunities [1, 5, 18]. Developing a wide range of foundational movement skills (e.g., catching, jumping, swimming, cycling) supports children to engage in a wide range of PAs [27, 28]. However, movement skills do not develop by maturation alone, children need to be physically active within favourable conditions for movement skills to emerge and progress, such as through structured teaching and learning activities [29]. The

more a child moves the greater the opportunity to develop and acquire competence in movement skills [30, 31], which should lead to enhanced engagement in PA [27, 30, 31]. Therefore, from a PE perspective, pedagogical approaches aimed at fostering movement competence should also seek to maximise opportunities for students to be physically active.

Pedagogical models designed for movement development can be beneficial for teachers as they provide a task structure so students can achieve intended learning outcomes [32–35]. Linear and Nonlinear pedagogy are two pedagogical approaches underpinned by different theories of motor learning that can guide the design of PE lessons aiming to foster the development of movement competence. Linear pedagogy is based on the Information Processing learning theory [36] and, in this perspective, a learner is seen as a system that elaborates perceptual-motor inputs to produce movement outputs [29]. Furthermore, learners participate in a set of planned movement experiences of increasing difficulty to obtain specific learning outcomes [29]. A central aspect of Linear pedagogy is to prioritise learning in the psychomotor domain through the repetition of movement tasks as repetition leads to movement automatization and therefore to increased accuracy and decreased cognitive load while performing the practiced task [37, 38]. Therefore, a key role of the educator is to design activities and provide instructions that are appropriate for children's proficiency level [37, 38]. Accordingly, Linear pedagogy is characterised by a teacher-centred approach to PE, where the teacher is the main source of instructional content and leads the performers through a series of pre-determined learning outcomes [29, 35]. In line with its theoretical foundation, Linear pedagogy includes the following characteristics: a) children should demonstrate mastery of the teacher-led movement patterns; b) movement skills should be broken down into simpler movements to facilitate movement proficiency; c) movement variability within a task is seen as detrimental for learning and therefore should be reduced [35, 38]. Interventions presenting Linear pedagogy characteristics were found to be effective at improving movement competence in children and adolescents [39–42].

Nonlinear pedagogy is based on ecological dynamics theoretical and philosophical foundations [43, 44]. From an ecological dynamics perspective learners are seen as complex neurobiological systems in mutual and reciprocal synergy with the environment that learn through perception and action coupling processes [34, 43, 44]. More specifically, perception and action coupling (or information-movement coupling) processes consist in the continuous creation of functional affordances (opportunities for action) within a cyclical process of perception and action leading to the emergence of goal-directed behaviours [45]. Based on this approach, learners are invited to explore different movement solutions within carefully designed learning environments. Proponents of this theory argue that it is the invitation for actions that leads to a continuous process of perceptual action coupling between the individual and the environment for intended movement solutions [34, 46]. Consequently, Nonlinear pedagogy is reported as a learner-centred PE approach where children are provided with high levels of autonomy and are invited to explore different movement solutions, while educators channel learning by modifying constraints [47]. Assumptions of Nonlinear pedagogy include the following: a) movement skills should be practiced in a situation that is representative of a game environment or performance condition, b) movement skills should emerge by the interaction between individual and environment in a movement perception action coupling: c) teachers modify individual, task and environmental constraints to channel movement skills learning; d) functional movement variability is encouraged; e) teachers should foster an external focus of attention [47, 48]. Recent studies showed that interventions following Nonlinear pedagogy principles can lead to improvement in movement competence within children and adolescents [49–51].

In summary, determining MVPA levels of children in PE and examining associated teaching practices can provide important information to assess adherence to guidelines associated with high quality PE. Movement competence is a key outcome for primary school PE and a feature of meaningful PE experiences for children. As movement development emerges through PA, our aim was to examine MVPA promotion within PE that use pedagogical approaches focused on movement competence. Our research could inform strategies to maximise meaningful opportunities to be physically active within PE lessons taught through these pedagogies. To date, no study has examined children's MVPA and teaching practices during PE using Linear and Nonlinear pedagogical approaches. Furthermore, no study has evaluated whether Linear and Nonlinear pedagogy would be associated with higher levels of children's MVPA and PA promoting teaching practices, compared to current PE. Therefore, this study aimed to assess children's MVPA and teaching practices in primary PE within Linear pedagogy and Nonlinear pedagogy and to compare this to current practice within PE delivery in primary schools.

## Materials and methods

### Design

This study was approved by the University Liverpool John Moores Research Ethics Committee (Reference 17/SPS/031) and formed part of the process evaluation of the Skill Acquisition Methods fostering Physical Literacy in Early Primary Education (SAMPLE-PE) cluster randomised controlled trial (ClinicalTrials.gov identifier: NCT03551366), which is described in detail elsewhere [52]. Specifically, this study was designed to evaluate the implementation of the interventions and explore PA promoting teaching practices during PE lessons and participants' responsiveness (that concerns the measurement of how far participants respond to, or are engaged by, an intervention [53]) in terms of children's MVPA engagement, rather than to evaluate changes in these constructs from baseline to post-intervention. Briefly, SAMPLE-PE aimed to investigate the efficacy of PE curricula based upon different pedagogical principles and motor learning theories in promoting physical literacy amongst 5-6-year-old children. One hundred and nineteen primary schools situated in deprived areas of a large metropolitan city in North West England were invited to take part in the study. Head-teachers from 12 primary schools provided gatekeeper consent and written parental consent and child assent were obtained for 360 5–6-year-old children (55% girls) from year 1 classes to participate in the research. Children without informed consent continued to participate in the PE lessons as normal. Children who were not able take part in PE due to reasons such as medical conditions, profound learning disabilities or special educational needs were not eligible to take part in this study. Using a computer-generated procedure, schools were randomly allocated to one of three groups: i) Nonlinear pedagogy PE intervention ($n$ = 3 schools); ii) Linear pedagogy PE intervention ($n$ = 3 schools); or iii) control group ($n$ = 6 schools). Following baseline assessments, intervention schools received a 15-week PE curriculum intervention delivered by trained coaches, while control schools followed usual practice (described in detail below). All groups were asked to provide the same dose of PE (i.e., 2 × 60 min weekly PE lessons, for 15 weeks).

Outcome data were collected at baseline (T0), immediately post-intervention (T1), and 6 months after the intervention has finished (T2). The process evaluation methods have been published in the study protocol [52], and only relevant methods for the current study analyses are outlined below. For feasibility and time constraint reasons and in line with sample size calculations reported below, a convenience sample of 50% of the children who provided consent to participate in the SAMPLE-PE project within 9 schools (comprising 3 Nonlinear

intervention schools, 3 Linear intervention schools and 3 randomly selected control schools) were recruited for this study.

## Sample size and statistical power

Sample size and power calculations for the SAMPLE-PE cluster-randomised controlled trial are reported elsewhere [52]. For the purposes of this study, an a priory power calculation was undertaken using G*Power software to detect differences between 3 groups including a large effect size based on the review by Fairclough and Stratton [54], 90% power, alpha levels set at p < 0.05 and multiple covariates recommended a minimal sample size of 83 children. It was not possible to account for clustering factors (e.g. school) in the power calculation as the mixed model analysis reported in previous literature did not report ICCs associated with clustering factors. Previous studies that have assessed MVPA during PE included a sample size similar or higher than 83 children (e.g. up to 168 children) [55–60]. Therefore, in line with the power calculation and the sample sizes observed in previous research, and after accounting for potential dropout, we aimed to recruit 50% of the research participants, amounting to 157 children, which was considered adequate for the purpose of this study [52]. Due to the lack of previous research reporting effect sizes about SOFIT+ outcomes and feasibility factors such as time and resource constraints and school burden, we aimed to collect data about teaching practices in 3 lessons per class participating in the project.

## Intervention

Intervention deliverers were recruited and trained to deliver Linear or Nonlinear pedagogy interventions [52]. Both Linear and Nonlinear pedagogy PE curricula were delivered over 2 lessons a week for 15 weeks leading to a total of 30 PE lessons per class divided into 3 content blocks of 10 lessons (each block lasting 5 weeks) focusing sequentially on dance, gymnastics and then ball skills, respectively. Teachers and sport coaches within control schools delivered PE as usual 2 lessons a week for 15 weeks.

## Deliverer training and intervention delivery

Intervention deliverers were recruited from a University in the North-West of England with a longstanding reputation for delivering high quality BA (Hons) Physical Education and BSc (Hons) Sport Coaching undergraduate and postgraduate degree programmes. As a result, two sport coaches from the research team and three sport coaches who each possessed at least a level 2 coaching qualification, were recruited and agreed to participate in a series of training sessions, to support the delivery of the SAMPLE-PE interventions. Before commencing the training, each one of the coaches was observed by a member of the research team while delivering a PE lesson in a primary school not involved in the SAMPLE-PE project. The coaches were then assigned to either a Linear (n = 2) or Nonlinear (n = 3) curriculum training programme based on their observed pedagogical approaches. The training for each pedagogy was designed to incorporate both practical and theoretical elements and was delivered by members of the research team with expertise in these approaches. Each training session lasted approximately 180 minutes and was conducted over a period of five weeks. During the training programme the coaches had the opportunity to be observed leading a PE lesson with Year 2 children (6-7-years-old) within a primary school not participating in the SAMPLE-PE project. Following these lessons, the coaches received augmented feedback from members of the research team. They were also encouraged to reflect on their pedagogic practice and encouraged to develop strategies to improve their own self-analysis. Following the training period coaches received a pedagogical framework and a resource pack together with the material used

during the sessions and recordings of the practical sessions. The PE lessons were planned considering equipment available or that could be made available in each one of the participating schools.

## Linear pedagogy intervention delivery

Linear pedagogy PE lessons were designed following the principles of Information Processing theory, informed by concepts of direct instruction [35], and followed a task structure involving: 1) a teacher-led warm-up activity, 2) practicing movement skills within drills, 3) a performance or game activity to apply the movement skills learnt during the lesson 4) a cool down (S1 Table). The coaches were expected to plan learning tasks and provide clear verbal instructions and visual demonstrations to provide the children with a 'picture' of what proficient movement looked like. During early learning of a movement skill the coaches were encouraged to review previously learned material and to provide corrective feedback during each activity with particular attention to children reiterating mistakes. The coaches were trained to use Fitts and Poster's cognitive stages (cognitive, associative, autonomous) [38] to evaluate children's progression in movement skills proficiency and to change the difficulty of the tasks based on children's skill level. Children were invited to perform and repeat movement skills as previously demonstrated by the educators and once the skill showed signs of automaticity were encouraged to practice independently in increasingly open environments. Gentile's taxonomy principles together with the Challenge Point framework [61, 62] were used by the teachers to facilitate these progressions of skill practice into more open environments.

## Nonlinear pedagogy intervention delivery

The Nonlinear pedagogy intervention was designed in line with an ecological dynamics framework [34]. For instance, each PE lesson started with children exploring the PE hall and different equipment within the environment (e.g., benches, mats, hoops, cones). The lesson continued with activities where teachers introduced variability by changing constraints and tasks designed to be representative of a real game, sport or performance. The children were invited to explore opportunities for action (affordances) and encouraged to create functional movement solutions (S2 Table). Educators were asked to use the Space, Task, Equipment, People (STEP) framework to identify and modify constraints within the lessons [63]. Furthermore, coaches were trained to use Newell's stages of motor learning (coordination, control and skill) to monitor children's progress in movement learning and to modify and individualise constraints based on the motor learning stages observed [64]. Demonstrations or corrective feedback were not used during activities, alternatively, coaches invited children to observe their peers in action, or prompted children to try to find different movement solutions (increasing exploration). Coaches were encouraged to use dialogue as a strategy to foster an external focus of attention in the child to infuse variability in the task and channel children learning (e.g., how can you make a pass that is easier to catch for your teammate? How many ways to move on the mat can you find?).

## Measures and procedures

Child anthropometric and demographic data were collected at schools during baseline assessments (between January and February 2018), within a two-week period before the start of the intervention. Children's PA levels (accelerometers), teaching practices related to PA (video observation) and pedagogical fidelity (video observation) were assessed during PE lessons as part of the SAMPLE-PE process evaluation between February and June 2018 [52]. Specifically, three PE lessons in each year 1 class (1 lesson every 5 weeks) were randomly selected for data

collection. Each of the intervention groups and the control group included five Year 1 classes. Therefore, 45 lessons (15 per group) were scheduled to be evaluated. Schools were informed about the data collection schedule before the beginning of the trial.

## Anthropometrics

Body mass was assessed to the nearest 0.1 kg using scales (model 760, Seca, Hamburg, Germany) while stature was assessed using stadiometers to the nearest 0.1 cm (The Leicester Height Measure, Child Growth Foundation, Leicester, United Kingdom) [65]. All anthropometric measurements were taken twice while a third measurement was taken in case the first two measurements differed by more than 1% and subsequently the mean between the measurements was taken. Body mass index (BMI) was calculated using stature and mass measurement and then it was converted to standardised BMI z-scores following international Obesity task force (IOTF) classification [66].

## Demographics

Children's demographic data (i.e., date of birth, sex, ethnicity, household postcode) were collected using questionnaires that parents filled and returned together with the consent form. Children's neighbourhood deprivation rank and decile were calculated from household postcode using the English indices of deprivation [67].

## Physical activity measurement

ActiGraph GT9X (ActiGraph, Pensacola, FL, USA) were used to assess PA in children during PE. Before the beginning of each lesson, ActiGraph GT9X accelerometers were fitted on each participant's non-dominant wrist to assess their PA levels during the lesson. If one of the randomly selected children was absent another participant to the SAMPLE-PE project was randomly selected to wear an accelerometer. Accelerometers were set to record accelerations at 100Hz over 1 second epochs within a range of ±8 g on x, y and z axes. Raw acceleration data were downloaded from accelerometers in 1 s epochs and exported as .csv files using ActiLife software (ActiGraph, Pensacola, FL, USA). Raw data were then transformed into Euclidean Norm Minus One (ENMO) acceleration data using GGIR package [68] from R software Version 4.0.2 (www.r-project.org). Lastly, age appropriate cut-points by Crotti et al. (2020) were used to classify ENMO accelerations equal or higher than 189 m$g$ into time spent in MVPA [69].

## Teaching practices related with physical activity: SOFIT+

PE video-recordings were analysed by one researcher using the modified version of the System for Observing Fitness Instruction Time to measure teacher practices related with PA (SOFIT+) [70]. SOFIT+ is a valid and reliable observation tool designed to classify multiple teaching practices related with children's PA during PE [70]. The teaching practices within the SOFIT + are divided in 4 categories comprising *Lesson Context*, *Activity Context*, *Teacher Behaviours* and *Activity Management* and more information about the definition of each teaching practice can be found in S3 Table. Each SOFIT+ scan lasts 40 seconds divided in two 20 seconds phases each one comprising 10 seconds of observation and 10 seconds of recording [70]. During the phase 1 of SOFIT+, *Lesson Context* and *Activity Context* teaching practices are assessed while during phase 2 *Teacher Behaviours* and *Activity Management* are assessed [70].

## Fidelity

Intervention fidelity in terms of Linear and Nonlinear pedagogy were independently assessed through the video analysis of recorded PE lessons using a checklist developed by the research team (S4 Table) [71]. The checklist comprised 9 items including 7 motor learning related items and 2 global items. Each item was rated using a 1 to 5 Likert sale where values of 1 and 2 corresponded to the observation being in line with Linear pedagogy while values of 4 and 5 corresponded to the observation being in line with Nonlinear pedagogy. Motor learning related items were assessed 4 times within each lesson (once for each quartile of the PE lessons) while global items were assessed only once per lesson observed. Two researchers that were not part of the research team and that were blinded to intervention allocation independently coded the fidelity of the PE lessons following training. The training consisted in 1) reading specific literature concerning Linear and Nonlinear pedagogy, 2) reading the fidelity checklist, 3) consulting the research team about doubts concerning the checklist, 4) independently coding 2 PE lessons, 5) consulting a pedagogy expert to check the coded lessons and clarify any doubts, 6) collaborating to assess 6 PE lessons, 7) independently assessing 6 lessons and then compare the results. The coders then assessed fidelity using the fidelity checklist within a total of 13 randomly selected PE lessons from Linear pedagogy (5 lessons), Nonlinear pedagogy (5 lessons) and control group (3 lessons).

## Data analysis

All data analysis was carried out using R Software (Version 4.0.2, www.r-project.org) and RStudio Software (Version 1.3.1056, www.rstudio.com). Multilevel models were used to analyse PA outcomes to account for MVPA data (level 1) being nested within child (level 2), class and teacher (level 3). Multilevel models were fitted using "Lme4" package [72]. To assess the association between pedagogy and MVPA during PE, two models were designed with children's MVPA during PE as the dependent variable: i) an unadjusted model including group (i.e., Linear pedagogy, Nonlinear pedagogy and control) as the independent variable with data nested by child (random intercept), and ii) a fully adjusted model including group (i.e., Linear pedagogy, Nonlinear pedagogy and control) as the independent variable and controlling for sex [14], age [14], lesson duration [16], lesson content (e.g., ball games) [14], lesson environment (i.e., indoor, outdoor) [21] with child id code, school and teacher included as nesting variables. During the modelling process, we decided to include variables that significantly increased the fit of the model and to exclude the nesting level of school class as it did not lead to an improved model fit or led to overfitted models. IOTF BMI z-score, ethnicity and deprivation decile variables were excluded from the fully adjusted multilevel analysis as they did not improve model fit and led to issues with listwise deletion of missing data and the loss of 21 participants and 50 corresponding valid MVPA observations within the multilevel models. The unadjusted and fully adjusted models were fitted using control group or Nonlinear pedagogy group as the 'group' reference category to evaluate whether Linear and Nonlinear interventions were associated with increased or decreased MVPA minutes or percentage of MVPA (MVPA%) compared to the control group and each other. Outliers were identified using absolute deviation around the median [73] and then removed from the dataset used for the final analysis.

It was not possible to use multilevel models to analyse the PA teaching practices data as most teaching practices variables did not present a normal distribution of the residuals or led to overfitting problems within the multilevel models. PA teaching practices observations collected using SOFIT+ are count data (representing counts of events over a discrete time span) [74–76]. Therefore, Poisson and Negative Binomial were initially considered for data analysis.

The dispersion of the data was assessed using Dean's test [77]. Given that all the distributions of teaching practice data were over-dispersed, Negative binomials were used to evaluate differences in PA teaching practices between Linear pedagogy, Nonlinear pedagogy and control group within PE. In some cases (i.e., *Partner Activity* and *Small Sided Activity*), negative binomial models could not fit the data as an elevated proportion of zero counts were observed. In these cases, hurdle negative binomial models were employed to analyse teaching practices data [74–76, 78]. To account for differences in lesson duration an offset factor was included in Negative binomial and Hurdle Negative binomial models. The statistical model fit of count data models were assessed using McFadden's pseudo R squared [79]. Due to the relatively small number of lessons observed within each group and for each PE deliverer, it was not possible to add covariates to the Negative binomial models as it was leading to overfitting (models failing to converge).

The datasets and related metadata (statistical analysis codes used to analyse the data) are publicly available within an open access repository (link to the data and metadata: https://doi.org/10.24377/LJMU.d.00000102).

## Results

Participants in the current study ($n$ = 162; 53% girls) presented a mean age of 6.0 (Standard Deviation (SD) = 0.3) years, 49% were white British, and 84% of the children lived in areas ranked as within the most deprived tertile for deprivation in the England. IOTF BMI z-scores were calculated for the 146 children and, based on IOTF thresholds [66], 24% of children were overweight or obese (Table 1). Parents did not report neighbourhood deprivation for 1 child in the control group, while ethnicity information was not provided for 2 children in the Linear pedagogy group and 2 children in the Nonlinear pedagogy group. Due to time constraints, we were not able to measure the BMI of 3 children from the Linear pedagogy group, 4 children from the Nonlinear pedagogy group and 9 from the Control group.

Each of the 15 participating classes were observed 3 times during PE. In total, 44 PE lessons were recorded as two classes within the control group did one PE lesson together. Audio was

**Table 1. Participants' descriptive data by group.**

| | Linear pedagogy | | Nonlinear pedagogy | | Control | |
|---|---|---|---|---|---|---|
| | (n = 55) | | (n = 65) | | (n = 42) | |
| | Mean (SD) or % | Missing data | Mean (SD) or % | Missing data | Mean (SD) or % | Missing data |
| **Decimal Age (years)** | 6.0 (0.3) | 0 | 5.9 (0.3) | 0 | 5.9 (0.3) | 0 |
| **Girls** | 56% | 0 | 49% | 0 | 55% | 0 |
| **White British** | 62% | 2 | 56% | 2 | 24% | 0 |
| **Living within the 30% most deprived areas** | 93% | 0 | 71% | 0 | 95% | 1 |
| **IOTF SDS BMI** | 0.4 (1.2) | 3 | 0.5 (1.1) | 4 | 0.2 (1.1) | 9 |
| **IOTF SDS BMI classification** | | | | | | |
| *Thinness grade 3* | 0% | | 0% | | 0% | |
| *Thinness grade 2* | 4% | | 2% | | 0% | |
| *Thinness grade 1* | 2% | | 3% | | 6% | |
| *Healthy weight* | 67% | | 75% | | 67% | |
| *Overweight* | 25% | | 8% | | 21% | |
| *Obese* | 2% | | 11% | | 6% | |

**IOTF SDS BMI:** standardised BMI z-scores following international Obesity task force classification.

not recorded in one of the control PE lessons because of technical problems. 43 PE lessons were assessed using SOFIT+ and combined with children's corresponding PA data for analyses. PA levels during PE were assessed in 42 (23 girls) children from the Control group, 65 (32 girls) children from the Nonlinear pedagogy group and 55 (31 girls) children from the Linear pedagogy group. Due to child absence from school, 114 (56 girls) children were assessed over 3 lessons, 32 (24 girls) children were assessed over 2 lessons, and 16 (6 girls) children were assessed over 1 lesson.

## Pedagogic fidelity

Pedagogic Fidelity scores were reported in Table 2. Nonlinear pedagogy average intervention fidelity scores ranged from 3.95 (SD = 0.78) to 5 (SD = 0.00), Linear pedagogy intervention average fidelity scores were all lower than 1.77 (0.94), while control group average scores were comprised between 1.44 (SD = 0.97) and 2.50 (SD = 0.54) [71]. Fidelity scores of 1 and 2 on the Likert scale correspond to the observation being more in line with Linear pedagogy and scores of 4 and 5 correspond to the observation being in line with Nonlinear pedagogy. Therefore, the fidelity observations indicated that Linear and Nonlinear interventions were delivered in line with their respective pedagogical principles. The control group presented fidelity scores indicated closer alignment with Linear pedagogy principles.

## Children's moderate to vigorous physical activity during physical education lessons

The mean and standard deviation for MVPA minutes, MVPA% and number of children spending 50% of PE time in MVPA can be found in Table 3. On average, children in the different groups engaged in MVPA during PE lessons for between 9.1 and 11.9 minutes, with the proportion of lesson time spent in MVPA ranging from 29.1% and 38.4%. The percentage of children engaging in MVPA over at least 50% of PE time ranged from 5.3% to 14.4% (Fig 1).

## Associations between pedagogy and children's physical activity

Results from the multilevel model analyses evaluating the associations between pedagogy group and children's average time spent in MVPA minutes during PE are reported in Table 4, while results concerning MVPA% during PE can be found in Table 5. Both Linear and Nonlinear interventions were associated with significantly higher minutes in MVPA and MVPA% percentage compared to the control group within the unadjusted models. However, within the fully adjusted models, Linear and Nonlinear pedagogy were not associated with increased MVPA or MVPA% compared to control group. Furthermore, Linear and Nonlinear pedagogy were not associated with higher MVPA or MVPA% compared to each other both in the unadjusted and fully adjusted model.

**Table 2. Pedagogical fidelity checklist results.**

|  | Category | | | | | | | Global | |
|---|---|---|---|---|---|---|---|---|---|
|  | Category Mean (SD) | | | | | | | Global Mean (SD) | |
|  | 1 | 2 | 3 | 4 | 5 | 6 | 7 | 1 | 2 |
| Nonlinear | 5.00 (0.00) | 5.00 (0.00) | 4.90 (0.28) | 3.95 (0.78) | 4.05 (0.77) | 4.73 (0.41) | 4.58 (0.43) | 5.00 (0.00) | 5.00 (0.00) |
| Linear | 1.40 (0.64) | 1.48 (0.85) | 1.20 (0.41) | 1.77 (0.94) | 1.20 (0.41) | 1.63 (0.88) | 1.63 (0.75) | 1.40 (0.74) | 1.33 (0.82) |
| Control | 2.10 (0.83) | 2.15 (1.04) | 2.19 (0.88) | 1.44 (0.97) | 2.33 (0.87) | 2.21 (0.75) | 2.50 (0.54) | 2.00 (1.08) | 1.92 (1.11) |

**SD:** standard deviation

**Table 3. Physical activity outcomes derived from accelerometers and teaching practices assessed using SOFIT+.**

| | Linear pedagogy | | Nonlinear pedagogy | | Control | |
|---|---|---|---|---|---|---|
| | Mean | SD | Mean | SD | Mean | SD |
| **Physical activity during PE** | | | | | | |
| MVPA (minutes) | 11.4 | 3.7 | 11.9 | 4.3 | 9.1 | 4.0 |
| MVPA (%) | 35.1 | 10.1 | 38.4 | 10.9 | 29.1 | 11.4 |
| Children spending ≥50% of PE time in MVPA (%) | 9.0 | 13.1 | 14.4 | 17.9 | 5.3 | 16.6 |
| ***SOFIT+ Lesson Context*** | | | | | | |
| Management (%) - | 23.9 | 7.7 | 22.2 | 9.2 | 40.2 | 17.2 |
| Knowledge (%) - | 25.5 | 12.6 | 14.9 | 9.9 | 22.5 | 8.3 |
| Motor Content (%) + | 50.6 | 10.5 | 62.8 | 14.7 | 37.3 | 15.1 |
| Fitness (%) + | 2.7 | 4.9 | 0.2 | 0.9 | 2 | 4.8 |
| Skill Practice (%) + | 45.1 | 9.7 | 0.6 | 2 | 17.2 | 22.6 |
| Game Play (%) + | 2.7 | 4.3 | 21.2 | 34.8 | 18.1 | 12.7 |
| Free Play (%) - | 0 | 0 | 0 | 0 | 0 | 0 |
| Discovery Practice (%) + | 0.1 | 0.4 | 40.8 | 27.8 | 0 | 0 |
| ***SOFIT+ Activity Context*** | | | | | | |
| Individual Activity (%) + | 25.9 | 16.1 | 24.3 | 20.3 | 4.7 | 12.8 |
| Partner Activity (%) + | 14.8 | 16.7 | 13.6 | 25.1 | 14.9 | 21.6 |
| Small Sided Activity (%) + | 4.5 | 8.6 | 3.7 | 8.3 | 3.8 | 9.3 |
| Large Sided Activity (%) - | 0 | 0 | 15.9 | 32.9 | 2.2 | 5.5 |
| Whole Class Activity (%) + | 5.4 | 6.2 | 5.3 | 10.6 | 11.7 | 12.6 |
| Waiting Activity (%) - | 9.5 | 11.1 | 0.3 | 0.8 | 7.9 | 13.2 |
| Elimination Activity (%) - | 0 | 0 | 0 | 0 | 3.5 | 8.6 |
| Girls Only Activity (%) - | 0 | 0 | 0 | 0 | 0 | 0 |
| Children Off Task (%) - | 6.8 | 7.1 | 6.6 | 6.2 | 2 | 2.7 |
| ***SOFIT+ Teaching Behaviours*** | | | | | | |
| Supervises (%) + | 24.3 | 8 | 16.6 | 11.9 | 20.7 | 15.1 |
| Instructs Single Child (%) - | 17.7 | 11.3 | 31.7 | 14.7 | 27.1 | 12.9 |
| Instructs Group (%) - | 6.4 | 6.7 | 24.7 | 17.8 | 7.7 | 7.8 |
| Instructs Class (%) - | 41 | 14.1 | 26.5 | 13.7 | 38.5 | 11.2 |
| Promotes PA (%) + | 0 | 0 | 0.2 | 0.6 | 0.6 | 1.6 |
| PA as Punishment (%) + | 0 | 0 | 0 | 0 | 0 | 0 |
| Withholding PA (%) - | 0.1 | 0.4 | 1.4 | 5.5 | 0.9 | 3.3 |
| PA Engaged (%) + | 8 | 6 | 0 | 0 | 3 | 4.4 |
| Off Task (%) - | 2.6 | 2.8 | 0.5 | 0.9 | 3 | 2.6 |
| ***SOFIT+ Activity Management*** | | | | | | |
| Signalling (%) - | 5.9 | 4.5 | 4.7 | 4.6 | 3.1 | 2.6 |
| Retrieving equipment M* (%) - | 0 | 0 | 0 | 0 | 0 | 0 |
| Retrieving equipment O* (%) - | 1.3 | 2.1 | 0.3 | 0.7 | 1.7 | 2.6 |
| Interruption Public (%) - | 3.8 | 2.4 | 4.7 | 3.7 | 5.6 | 5.6 |
| Interruption Private (%) - | 1.5 | 1.8 | 6 | 4.5 | 4.6 | 4.2 |

**SD:** standard deviation; **PE:** physical education; **M***: multiple points; **O***: one point; **+:** the teaching practice was theorised to foster children's moderate to vigorous physical activity; **-:** the teacher practice was theorised to reduce children's moderate to vigorous physical activity.

Within the fully adjusted models, sex was significantly and negatively associated with both MVPA minutes and MVPA%, meaning that girls were generally less active than boys during PE. Age was not significantly associated with MVPA minutes and MVPA%. PE lessons

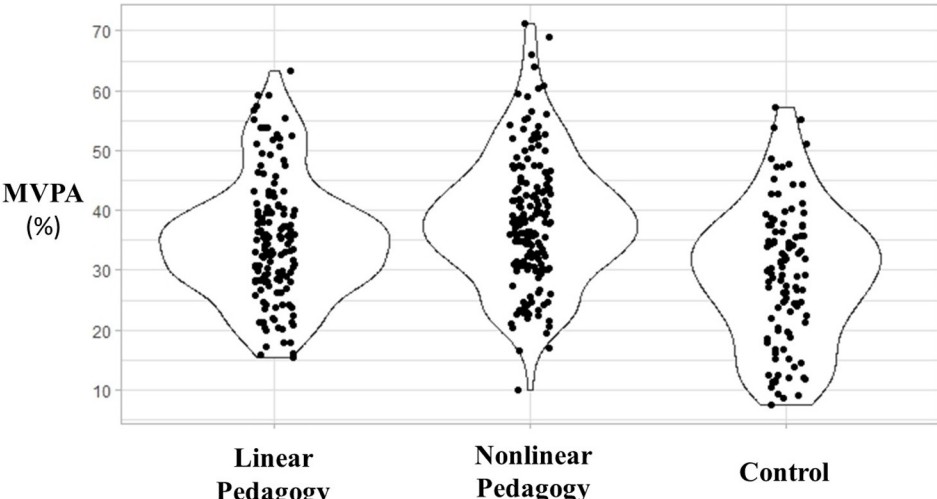

**Fig 1. Percentage of time spent in moderate to vigorous physical activity in physical education.** Fig 1 presents a violin density plots (shapes delimited by line) and dot plots concerning percentage of time spent in MVPA during PE; Each dot represents a single unadjusted MVPA measurement in one child during one lesson and dots were randomly scattered on the horizontal axis.

delivered outdoors were associated with higher MVPA minutes in children compared to lessons indoors. Ball games lesson content was found to be associated with higher MVPA minutes and

**Table 4. Association between pedagogy group and children's minutes of moderate to vigorous physical activity during physical education.**

| Predictors | Unadjusted model | | | Fully adjusted model | | |
|---|---|---|---|---|---|---|
| | Estimate | CI | p-value | Estimate | CI | p-value |
| Group [Nonlinear vs Control] | 2.58 | 1.56 – 3.60 | <0.001 | 1.54 | -2.45–5.53 | 0.450 |
| Group [Linear vs Control] | 2.37 | 1.32 – 3.41 | <0.001 | 0.73 | -3.58–5.04 | 0.740 |
| Group [Linear vs Nonlinear] | -0.21 | -1.15 – 0.72 | 0.652 | -0.81 | -3.18–1.56 | 0.503 |
| Sex | | | | -1.12 | -1.74 – -0.50 | <0.001 |
| Decimal Age | | | | 1.03 | -0.06 – 2.13 | 0.068 |
| Lesson Location | | | | 2.45 | 0.54 – 4.35 | 0.012 |
| Lesson content [Ball Games] | | | | 2.49 | 1.42 – 3.57 | <0.001 |
| Lesson content [Dance] | | | | 1.18 | -1.45 – 3.82 | 0.379 |
| Lesson content [Gymnastic] | | | | 2.65 | -0.14 – 5.45 | 0.063 |
| Lesson Duration | | | | 0.26 | 0.21 – 0.32 | <0.001 |
| $\sigma^2$ | 12.75 | | | 4.86 | | |
| $\tau_{00}$/ICC Participants | 1.76 | | | 1.83/0.14 | | |
| $\tau_{00}$/ICC Schools | | | | 1.04/0.08 | | |
| $\tau_{00}$/ICC Teachers | | | | 5.63/0.42 | | |
| ICC random factors | 0.12 | | | 0.64 | | |
| Participants | 162 | | | 162 | | |
| Schools | | | | 9 | | |
| Teachers | | | | 9 | | |
| PA Observations | 416 | | | 416 | | |
| Marginal $R^2$ / Conditional $R^2$ | 0.075 / 0.187 | | | 0.371 / 0.771 | | |

$\sigma^2$: Intercept variance; $\tau_{00}$: Random factor variance; **ICC:** intraclass correlation index; **PA:** physical activity.

**Table 5. Association between pedagogy group and children's percentage of moderate to vigorous physical activity during physical education.**

| Predictors | Unadjusted model | | | Fully adjusted model | | |
|---|---|---|---|---|---|---|
| | Estimate | CI | p-value | Estimate | CI | p-value |
| Group [Nonlinear vs Control] | 8.68 | 5.82 – 11.55 | <0.001 | 7.30 | -3.80–18.40 | 0.197 |
| Group [Linear vs Control] | 6.17 | 3.23 – 9.10 | <0.001 | 5.54 | -6.62–17.70 | 0.317 |
| Group [Linear vs Nonlinear] | -2.52 | -5.14 – 0.11 | 0.060 | -1.76 | -8.21–4.69 | 0.594 |
| Sex | | | | -3.60 | -5.57 – -1.64 | <0.001 |
| Decimal Age | | | | 2.99 | -0.50–6.47 | 0.093 |
| Lesson Location | | | | 4.82 | -1.12–10.75 | 0.112 |
| Lesson content [Ball Games] | | | | 7.54 | 4.15 – 10.93 | <0.001 |
| Lesson content [Dance] | | | | -0.35 | 8.58–7.89 | 0.934 |
| Lesson content [Gymnastic] | | | | 6.81 | -1.90–15.51 | 0.126 |
| Lesson Duration | | | | -0.34 | -0.51 – -0.17 | <0.001 |
| $\sigma^2$ | 89.15 | | | 48.39 | | |
| $\tau_{00}$/ICC Participants | 18.09 | | | 18.44/0.16 | | |
| $\tau_{00}$/ICC Schools | | | | 5.29/0.04 | | |
| $\tau_{00}$/ICC Teachers | | | | 43.87/0.38 | | |
| ICC random factors | 0.17 | | | 0.58 | | |
| Participants | 162 | | | 162 | | |
| Schools | | | | 9 | | |
| Teachers | | | | 9 | | |
| PA Observations | 416 | | | 416 | | |
| Marginal $R^2$ / Conditional $R^2$ | 0.100 / 0.251 | | | 0.239 / 0.682 | | |

$\sigma^2$: Intercept variance; $\tau_{00}$: Random factor variance; **ICC:** intraclass correlation index; **PA:** physical activity.

MVPA% compared to locomotor activities (reference category). Lesson duration was significantly and positively associated with MVPA minutes and negatively associated with MVPA%.

## Teaching practices associated with physical activity

The characteristics of PE lessons in terms of lesson content, lesson duration, lesson location, and teacher delivery are reported in Table 6. PE lessons lasted 32:07 mins on average (SD = 06:14 mins) and 14 out of 44 lessons took place outdoors. The observed PE lessons were delivered by 4 teachers and external sports coaches in the control group while 5 trained sports coaches delivered the observed PE lessons between interventions as reported in Table 6. Due to the restricted availability of deliverers during the intervention period, the two coaches recruited from the research team delivered both Nonlinear pedagogy and Linear pedagogy as they were trained in both pedagogical approaches (Table 6).

The mean and standard deviation concerning teaching practices divided by group can be found in Table 3. Furthermore, Table 3 indicates whether the teacher practice was theorised to foster or to hinder children's engagement in MVPA during PE [70, 80, 81]. SOFIT+ teaching practice variables comprising *Free play*, *Girls Only activity*, *PA as Punishment* and *Retrieving equipment from multiple access points* were never observed during the PE lessons (Table 3), while *Withholding PA* and *Large Sided Activity* teaching practices were only observed in 3 and 6 lessons, respectively. Therefore, a statistical analysis could not be completed for these variables.

The results from the analysis of teaching practices can be found in Table 7. Regarding *Lesson Context* variables, Linear pedagogy included higher incidences of *Motor Content* and *Skill*

**Table 6. Lesson characteristics.**

| | Linear pedagogy | Nonlinear pedagogy | Control |
|---|---|---|---|
| Lesson duration mean ± SD (minutes) | 34.2 ± 6.6 | 30.8± 6.8 | 31.2 ± 5.0 |
| Lessons observed | 15 | 15 | 13 |
| Locomotor activities | | | 8 |
| Gymnastic | 5 | 5 | |
| Dance | 5 | 5 | |
| Ball games | 5 | 5 | 5 |
| *Number of Physical education lesson by deliverer* | | | |
| Deliverer 1 | | | 3 |
| Deliverer 2 | | | 3 |
| Deliverer 3 | | | 6 |
| Deliverer 4 | | | 1 |
| Deliverer 5 | | 3 | |
| Deliverer 6 | | 7 | |
| Deliverer 7 | 4 | 1 | |
| Deliverer 8 | 2 | 4 | |
| Deliverer 9 | 9 | | |

*Practice* as well as lower incidences of *Management* and *Game Play*, compared to the control group. Nonlinear pedagogy group included higher incidences of *Motor Content* and *Discovery Practice* together with lower incidences of *Knowledge*, *Management*, *Skill Practice*, compared to the control group. Additionally, Linear pedagogy group involved higher incidences of *Knowledge* and *Skill Practice* and lower *Motor Content*, *Game Play* and *Discovery Practice*, compared to Nonlinear pedagogy group.

For *Activity Context* variables, Linear pedagogy included higher incidences of *Individual Activity* and *Children Off Task* as well as lower incidence of *Elimination Activity*, compared to the control group. Furthermore, Nonlinear pedagogy group involved higher incidences of *Individual Activity* and *Children Off Task* together with lower incidences of *Waiting Activity* and *Elimination Activity*, compared to the control group. Lastly, Linear pedagogy group involved an increased incidence of *Waiting Activity* compared to the Nonlinear pedagogy group.

For Teaching Behaviours variables, Linear pedagogy included higher incidence of *PA Engaged* and lower incidence of *Instructs Single Child*, compared to the control group. Furthermore, Nonlinear pedagogy group involved higher incidence of *Instructs Group*, as well as lower incidences of *Instructs Class*, *PA Engaged* and *Off Task*, compared to the control group. Additionally, Linear pedagogy group involved increased *Instructs Class*, *PA Engaged* and *Off Task* together with lower *Instructs Single Child* and *Instructs Group* compared to Nonlinear pedagogy group.

As regards *Activity Management* Variables, Linear pedagogy included lower incidence of *Interruption Private* compared to control group and Nonlinear pedagogy group while no other significant differences were found.

## Discussion

This study aimed to evaluate and compare children's MVPA, and teaching practices associated with MVPA, during primary school PE within different PE pedagogical approaches (Linear and Nonlinear) and current practice in PE. The results suggest that primary PE

**Table 7. Difference in teaching practices between the interventions and control group.**

| Teaching practice | Linear vs Control | | | Nonlinear vs Control | | | Linear vs Nonlinear | | | |
|---|---|---|---|---|---|---|---|---|---|---|
| | Incidence | Std. Error | p-value | Incidence | Std. Error | p-value | Incidence | Std. Error | p-value | McFadden |
| *Lesson Content* | | | | | | | | | | |
| Knowledge | 1.14 | 0.23 | 0.513 | **0.66** | **0.14** | **0.049** | **1.74** | **0.36** | **0.007** | 0.039 |
| Management | **0.59** | **0.08** | **<0.001** | **0.54** | **0.08** | **<0.001** | 1.08 | 0.16 | 0.609 | 0.065 |
| Motor Content | **1.36** | **0.15** | **0.005** | **1.70** | **0.18** | **<0.001** | **0.80** | **0.08** | **0.020** | 0.114 |
| Fitness | 1.35 | 1.37 | 0.769 | 0.13 | 0.17 | 0.104 | 10.06 | 12.04 | 0.054 | 0.037 |
| Skill Practice | **2.62** | **1.18** | **0.033** | **0.03** | **0.02** | **<0.001** | **76.29** | **49.79** | **<0.001** | 0.725 |
| Game Play | **0.15** | **0.10** | **0.006** | 1.18 | 0.78 | 0.806 | **0.13** | **0.09** | **0.002** | 0.042 |
| *Activity context* | | | | | | | | | | |
| Individual Activity | **5.81** | **3.02** | **0.001** | **5.43** | **2.83** | **0.001** | 1.07 | 0.51 | 0.886 | 0.532 |
| Partner Activity | 0.71 | 0.29 | 0.399 | 0.75 | 0.31 | 0.496 | 0.94 | 0.35 | 0.877 | 0.020 |
| Small Sided Activity | 0.68 | 0.31 | 0.400 | 0.53 | 0.25 | 0.184 | 1.29 | 0.53 | 0.538 | 0.028 |
| Whole Class Activity | 0.45 | 0.26 | 0.162 | 0.46 | 0.26 | 0.175 | 0.98 | 0.56 | 0.969 | 0.012 |
| Waiting Activity | 1.19 | 0.93 | 0.820 | **0.04** | **0.04** | **0.002** | **32.08** | **32.66** | **0.001** | 0.066 |
| Children Off Task | **3.74** | **1.91** | **0.010** | **3.48** | **1.78** | **0.015** | 1.08 | 0.46 | 0.866 | 0.054 |
| *Teaching Practices* | | | | | | | | | | |
| Supervises | 1.16 | 0.25 | 0.483 | 0.79 | 0.18 | 0.292 | 1.48 | 0.32 | 0.068 | 0.029 |
| Instructs Single Child | **0.66** | **0.13** | **0.038** | 1.17 | 0.22 | 0.404 | **0.57** | **0.11** | **0.003** | 0.040 |
| Instructs Group | 0.86 | 0.31 | 0.668 | **3.22** | **1.11** | **0.001** | **0.27** | **0.09** | **<0.001** | 0.080 |
| Instructs Class | 1.06 | 0.16 | 0.694 | **0.68** | **0.11** | **0.015** | **1.56** | **0.24** | **0.003** | 0.032 |
| PA Engaged | **2.62** | **1.13** | **0.025** | | | | | | | 0.034 |
| Off Task | 0.92 | 0.30 | 0.791 | **0.18** | **0.10** | **0.003** | **4.95** | **2.74** | **0.004** | 0.150 |
| *Activity Management* | | | | | | | | | | |
| Signalling | 1.86 | 0.65 | 0.077 | 1.47 | 0.53 | 0.287 | 1.26 | 0.40 | 0.457 | 0.015 |
| Retrieving equipment O | 0.83 | 0.53 | 0.767 | 0.17 | 0.16 | 0.052 | 4.81 | 4.31 | 0.080 | 0.076 |
| Interruption Public | 0.70 | 0.23 | 0.285 | 0.85 | 0.28 | 0.614 | 0.82 | 0.28 | 0.563 | -0.010 |
| Interruption Private | **0.31** | **0.12** | **0.003** | 1.24 | 0.39 | 0.484 | **0.25** | **0.10** | **<0.001** | 0.076 |

Significant results (p-value<0.05) were highlighted using bold font; **O:** One access point

interventions focusing on movement competence guided by Linear pedagogy and Nonlinear pedagogy were not associated with different levels of children's MVPA during PE when compared to current practice in PE. Other factors were associated with children's MVPA time and MVPA% in PE including the sex of the participants (boys), lesson duration (longer), lesson location (outdoors), lesson content (ball skills, gymnastic, dance), while the teacher providing the lesson also explained a high proportion of MVPA variance. Furthermore, only a small proportion of children engaged in MVPA for at least 50% of PE time both in the intervention (Linear pedagogy: 9.0%, Nonlinear pedagogy: 14.4%) and control groups (5.3%). As for teaching practices during PE, higher incidences of PA promoting teaching practices (e.g., *Motor Content*, *Skill Practice*, *Discovery Practice*, *Individual PA*, *PA Engaged*) and lower incidences of PA decreasing teaching practices (e.g., *Knowledge*, *Management*, *Instructs Class*, *Off Task*) were found in PE lessons guided by Linear and Nonlinear pedagogical approaches. Lastly, both Linear and Nonlinear interventions were delivered with high fidelity to the respective Linear and Nonlinear pedagogical principles. The results obtained in this study extend knowledge about MVPA promotion in early primary PE under different pedagogies.

## Increasing physical activity in physical education

As shown in Fig 1, the majority of children's MVPA levels within both intervention and control groups did not reach the recommended MVPA engagement of 50% of the PE lesson duration [4, 12, 13]. This is in line with the vast majority of studies assessing MVPA in PE using accelerometers and observation tools, even when those PE lessons were led by PE specialists whose aim was to promote high MVPA during PE [21, 23–25]. This suggests that high quality PE targeting other learning outcomes such as movement competence does not necessarily lead to specific thresholds of MVPA engagement. Therefore, future studies should seek to identify additional ways to promote PA whilst providing rich movement competence learning experiences for children.

This study was the first to evaluate the association between Linear pedagogy and Nonlinear pedagogy with children's MVPA and to compare PA engagement in these pedagogies with current practice in PE in primary schools. The results from this study suggest that Linear pedagogy or Nonlinear pedagogy was not a significant predictor of MVPA engagement in PE. The lack of an association between participation in the motor learning pedagogy interventions and children's MVPA in PE could be due to the intervention being designed to improve movement competence in children rather than MVPA [52]. Indeed, the vast majority of previous studies where higher levels of MVPA during PE were observed in the intervention group compared to the control condition included specific strategies to improve MVPA during PE (e.g., teacher training to deliver specific MVPA promoting PE content) and reported MVPA engagement during PE as being the primary outcome of the intervention [17, 25, 56–60, 82–84]. However, results from many of these previous studies should be interpreted with caution as, unlike the present study, they did not account for factors associated with MVPA in PE such as children's sex, age and BMI, lesson content, lesson location and lesson duration [31, 59, 60, 82–84] and/ or studies did not account for children being nested within schools, classes or teacher within their statistical analyses [59, 60]. Furthermore, of the studies assessing PA in PE, our study was the first reporting the pedagogical basis guiding the delivery of movement learning activities. As an example, the "Move it Groove it" and "PLUNGE" interventions reported both PA and movement skills development as aims of their PE interventions [55, 85]. However, despite describing strategies to improve MVPA in PE, neither of these two studies clarified the pedagogical basis guiding the delivery of movement learning activities [55, 85]. Therefore, we suggest that future research should further investigate how different pedagogies and PA promotion strategies might affect children's PA during PE. Furthermore, we recommend that clear descriptions of pedagogies and PA promotion strategies should be reported in future PE interventions studies as this could help both practitioners and researchers understanding how to achieve and/or prioritise specific PE outcomes (e.g. children's motor competence development or high MVPA engagement).

Although presenting different research design and aims compared to our study, lessons can be learned from some of the aforementioned primary school PE interventions that targeted the improvement of MVPA and PA promoting teaching practices and measured changes in these outcomes from baseline to post-intervention [17, 58, 59]. Based on the findings from the Partnerships for Active Children in Elementary Schools (PACES) intervention study [17] we suggest that future Linear and Nonlinear pedagogy interventions aiming to improve children's MVPA during PE could seek to increase *Small Sided Activity* as well as teacher *Promotes PA* time and reduce *Children off task* (i.e., time when one or more students are not engaged in the task proposed by the teacher). Furthermore, considering evidence from a follow-up to the PACES study by Weaver et al. (2018) [58] we advise that decreasing *Knowledge* time and increasing *Motor Content* time in future Linear and Nonlinear pedagogy interventions as well

as decreasing *Waiting Activity* in future Linear pedagogy interventions could also be effective and feasible strategies to foster children's MVPA in PE. Finally, the "SHARP" intervention [59] reported a significant increase in MVPA together with increased time in teaching practices such as *Skill Practice* and "in class PA promotion" within the intervention group compared to the control group. The increase in *Skill Practice* observed in the SHARP intervention could be associated with the SHARP principle concerning "high repetition of motor skills" that is also a key principle within the Linear pedagogical intervention delivered in this study suggesting that practicing movement skills can significantly contribute to MVPA in PE [59, 86]. Furthermore, the high percentages of verbal PA promotion within the SHARP (42.3%) intervention compared to that observed in this study (0–0.2%) confirms that future Linear and Nonlinear interventions could focus on improving verbal PA promotion during PE delivery as a strategy to improve children's MVPA in PE [59].

## Factors associated with children's physical activity in physical education

The teacher delivering PE explained a high proportion of variance in the fully adjusted models examining children's MVPA minutes (ICC = 0.42) and MVPA% (ICC = 0.37) [87] (Tables 4 and 5), suggesting that teachers are an important predictor of activity levels. More specifically, the high proportion of variance explained by the teachers in our models suggests that children doing PE with the same teacher reached similar levels of MVPA engagement during PA [87, 88]. In other words, some teachers were more effective in promoting MVPA in PE than others irrespective of them being in the intervention or in the control group. This could be due to the teacher's expertise and their knowledge and experience about strategies to engage children in high levels of PA [21, 23–25]. In line with this, PE lessons within the control group were delivered by a class teacher, two coaches (sports coaches hired from external sport coaching organisations), and a PE specialist teacher. This potentially explains why the mean MVPA and MVPA% observed in the control group (9.1 min, 29.1%) was similar or higher than previous studies in which PE was provided by generalist class teacher and reported levels of MVPA during PE ranged from 3.5 min to 10.8 min and MVPA mean percentage ranged 9.5% to 29.7% [14, 15, 89]. Interestingly, the mean MVPA percentages observed in the Linear (35.1%), and Nonlinear (38.4%) intervention groups were similar to the proportion of children's MVPA during PE observed in a study involving specialised PE teachers, with 36.7% of the lessons spent in MVPA [16]. This might be due to the intervention deliverers in the present study having experience in PE delivery in primary school children and to the intervention delivery not including generalist classroom teachers or it might be due to the content of the Linear and Nonlinear pedagogy interventions [21, 23–25].

Consistent with previous literature, it was found that MVPA during PE was associated with several factors with girls engaging in lower levels of MVPA and MVPA% compared with boys [14], longer PE lessons associated with higher minutes spent in MVPA but lower MVPA% [16], lesson content being associated with MVPA and MVPA% with ball games activities led to the highest MVPA and MVPA% engagement [14], and lastly, outdoor lessons being associated with higher levels of MVPA compared to indoor lessons when factoring teachers into the models [90]. In view of these results, researchers and practitioners should account for these factors when designing interventions to foster MVPA in PE. In particular, key aspects to consider should be: 1) finding strategies to engage girls in MVPA, for example, proposing activities that are meaningful and enjoyable for them [91]; 2) including relevant high intensity game activities with the PE lesson [14, 15]; 3) using outdoor spaces when the weather conditions allow as outdoor PE is associated with higher MVPA levels in children compared to indoor PE

[21], and 4) finding strategies to maximise lesson duration (e.g. making sure that the lesson starts and ends as established by the school curriculum) [16].

## Teaching practices in pedagogies underpinned by movement learning theories

The SOFIT+ data provided valuable information about the characteristics of Linear and Nonlinear pedagogy approaches in terms of teaching practices, which can be used to improve PE delivery to promote MVPA engagement in the future.

As expected from a teacher-centred pedagogical approach, the Linear pedagogy intervention involved higher *Skill Practice* and less *Game Play* compared to the Nonlinear pedagogy and control groups, as well as higher *Individual Activity* compared to the control group [35, 38, 71]. Furthermore, Linear pedagogy intervention involved a higher proportion of *Instructs Single Child* compared with other groups, and a higher proportion of instructing the class compared to the Nonlinear group in line with teacher-centred PE approaches [92, 93]. When compared to the control group, the Linear pedagogy intervention involved a higher proportion of time spent in *Motor Content* and teacher PA engagement that are associated with increased MVPA levels during PE together with less time spent in *Management* activities and *Elimination Activity* that are associated with decreased MVPA. However, within previous literature *Game Play* was found to be associated with the highest MVPA engagement in PE compared to other type of *Lesson Contexts* and within this study *Game Play* was observed less frequently in Linear intervention compared to the control group [14, 70, 80]. Furthermore, a higher percentage of *Children Off Task* was observed in Linear pedagogy group compared to control group. Therefore, future interventions guided by Linear pedagogy should consider increasing the proportion of time children spend in *Game Play* and find strategies to decrease *Children Off Task* within PE lessons to improve MVPA engagement.

As expected from a learner-centred pedagogical approach, the Nonlinear pedagogy intervention included a lower proportion of time in *Knowledge* and *Instructs Class* compared to other groups and it was practically the only intervention group where *Discovery Practice* was observed though *Skill Practice* was not [47, 48, 92, 93]. The lack of *Skill Practice* and the high proportion of *Game Play* is in line with the Nonlinear pedagogy principle of learning movement skills in a representative learning design [47, 48]. The Nonlinear intervention presented a higher proportion of MVPA promoting teaching practices (i.e. *Motor Content*) and a lower proportion of MVPA decreasing teaching practices (i.e. *Knowledge*, *Management*, *Waiting Activity*, *Elimination Activity*, *Instructs Class* and teacher being *Off Task*) compared to the control group. However, compared to the control group, the Nonlinear pedagogy intervention involved a higher proportion of *Children Off Task* (associated with decreased MVPA in PE) while teachers never engaged in PA with students, which is considered an MVPA promoting teaching practice. Therefore, future Nonlinear intervention should take in consideration aspects to decrease *Children Off Task* and for teachers to participate in PE as an active constraint to promote MVPA engagement. However, the lower levels of *Children Off Task* observed in the control group compared to both Linear and Nonlinear pedagogy could be due to teachers or coaches within the control group having a long relationship with the children leading to well established behavioral management strategies.

Lastly, both Linear and Nonlinear intervention presented none or almost no verbal promotion of PA engagement. This is likely due to these approaches not being focused on increasing MVPA engagement suggesting that this aspect could be improved in future interventions. Nevertheless, taking all the above findings together, the results suggest that Linear and Nonlinear pedagogical interventions both improve time allocated to movement competence practice

but would need to adopt more PA promoting teaching practices to increase children's MVPA in PE [70, 80, 81].

## Strengths and limitations

This study included several strengths comprising being the first study to analyse the association between Linear and Nonlinear pedagogy approaches in PE with children's MVPA in PE, and the first study to use accelerometery to report MVPA during PE among 5–6 years old children. A further strength was the simultaneous assessment of children's MVPA together with the observations of MVPA teaching practices by PE teachers within the same lessons. Another strength was that multilevel models accounting for different variables associated with children's MVPA were compared and that the models accounted for the nested structure of the data (i.e., observations being nested in children and children being nested in schools), while teaching practices data were analysed with the most appropriate models for count data. However, this study also has some limitations such as MVPA only being assessed in 50% of the children in the PE class that agreed to take part in the research project due to feasibility constraints. In relation, only 3 of the 6 control schools participating in the SAMPLE-PE project were included in this study. Furthermore, due to the relatively small amount of teaching practices data collected per group and per PE deliverer, it was not possible to account for factors such as teacher and lesson content in the teaching practice analysis and some teaching practices variables were only observed a few times, making it impossible to run a statistical analysis. Lastly, one PE lesson was excluded because of technical problems in the video recording of the lesson.

## Future directions

Future research could evaluate the implementation of movement learning pedagogical approaches in older children or adolescents to see if similar results are obtained compared to this study. Furthermore, future studies could include qualitative methods to examine children's PA experiences during PE under different pedagogical approaches and how experiences in PE within movement learning pedagogical approaches could affect children and young people's willingness to maintain high engagement in PE [94]. Future research assessing teaching practices associated with MVPA in PE should consider assessing a higher number of PE lessons per group and PE deliverers compared to this study with a particular attention to observe an adequate sample of PE lessons for each PE deliverer to collect teaching practices data allowing the design of complex statistical analysis models. Lastly, research could evaluate whether teacher professional training to deliver different pedagogies in PE as well as improving teaching practices associated with MVPA in PE might positively enhance their capacity and willingness to promote MVPA in PE sessions to improve movement competence.

## Conclusions

The majority of children's MVPA levels within both intervention and control groups did not reach the recommended MVPA engagement of 50% within PE in line with previous literature. Furthermore, compared to current practice in PE, interventions based on Linear and Nonlinear pedagogy were not associated with increased children's MVPA, but they included a higher incidence of MVPA promoting teaching practices (e.g., *Motor content*, *Skill Practice*, *Discovery Practice*). Nevertheless, the findings suggest that utilising Linear and Nonlinear pedagogies in PE could potentially improve movement competences in young children without compromising children's PA levels compared to general practice. Given that PE deliverers were the main predictor of MVPA in PE in this study, future interventions should focus on improving the pedagogic knowledge and skills of PE deliverers about increasing children's MVPA. This

paper provides valuable information about how teaching practices within different pedagogical approaches affect PA in PE and proposes teaching practices that should be targeted to improve MVPA in PE. These findings can be used to help practitioners and researchers who are interested in designing future PE or coaching interventions based on Linear or Nonlinear pedagogies and/or maximizing MVPA engagement in PE.

## Supporting information

**S1 Table. Linear pedagogy curriculum: Object control skills lesson.**
(DOCX)

**S2 Table. Nonlinear pedagogy curriculum: Invasion games lesson.**
(DOCX)

**S3 Table. Table reporting inter-rater reliability results and the definition of each teaching practice.**
(DOCX)

**S4 Table. Pedagogical fidelity checklist.**
(DOCX)

## Acknowledgments

The authors would like to thank Farid Bardid for his collaboration and valuable contribution in designing the SAMPLE-PE project. The authors would also like to thank Lynne Boddy for the support with PA measurement training and analysis. The authors also thank Kiersten Jones and Frederike Marie Stell for their help with the pedagogical fidelity check. Furthermore, the authors thank the children, classroom teachers and physical education deliverers for their participation in this study.

## Author Contributions

**Conceptualization:** Matteo Crotti, James Rudd, Simon Roberts, Katie Fitton Davies, Laura O'Callaghan, Till Utesch, Lawrence Foweather.

**Data curation:** Matteo Crotti, Till Utesch.

**Formal analysis:** Matteo Crotti.

**Investigation:** Matteo Crotti.

**Methodology:** Matteo Crotti.

**Project administration:** Matteo Crotti, James Rudd, Lawrence Foweather.

**Supervision:** James Rudd, Simon Roberts, Lawrence Foweather.

**Writing – original draft:** Matteo Crotti.

**Writing – review & editing:** Matteo Crotti, James Rudd, Simon Roberts, Katie Fitton Davies, Laura O'Callaghan, Till Utesch, Lawrence Foweather.

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
