## [Decision Letter · Decision Letter 0]

5 Apr 2022

PONE-D-21-38034Physical activity promoting teaching practices and children’s physical activity within physical education lessons underpinned by motor learning theory (SAMPLE-PE)PLOS ONE

Dear Dr. Foweather,

Thank you for submitting your manuscript to PLOS ONE. After careful consideration, we feel that it has merit but does not fully meet PLOS ONE’s publication criteria as it currently stands. Therefore, we invite you to submit a revised version of the manuscript that addresses the points raised during the review process. We had difficulty securing reviewers for this manuscript, hence the delay in getting to a first decision. In the interest of preventing any further delay, I reviewed the manuscript myself. I added my own comments, together with those from one reviewer, which I hope the authors would find useful. Please submit your revised manuscript by May 20 2022 11:59PM. If you will need more time than this to complete your revisions, please reply to this message or contact the journal office at plosone@plos.org. Please include the following items when submitting your revised manuscript:A rebuttal letter that responds to each point raised by the academic editor and reviewer(s). You should upload this letter as a separate file labeled 'Response to Reviewers'.A marked-up copy of your manuscript that highlights changes made to the original version. You should upload this as a separate file labeled 'Revised Manuscript with Track Changes'.An unmarked version of your revised paper without tracked changes. You should upload this as a separate file labeled 'Manuscript'.

We look forward to receiving your revised manuscript.

Kind regards,

Catherine M. Capio

Academic Editor

PLOS ONE

Journal Requirements:

Additional Editor Comments (if provided):

The study has potentially important contributions to the evidence related to physical education in relation to physical activity and motor competence of young children. There are areas in the manuscript that can be strengthened, which the authors might wish to consider:

- In the introduction, it was not clear why the researchers expected any differences in accrued PA between linear, non-linear, and usual practice conditions. While there linear and non-linear pedagogy were adequately explained, the underlying rationale for the research question was not made clear.

- Is there a sense of competition in PE for promoting motor competence vs. PA accrual? This was taken up in the discussion, but I believe this needs to be taken up earlier in the introduction.

- There was discussion of related studies such as SHARP, PACES - however, there was no critical discussion of how the current findings contribute/expand the relevant evidence base. I believe there is room for the authors to develop a more critical and insightful discussion to place their findings in the context of the wider evidence base.

- I wonder about the conclusion where the authors say that linear and non-linear pedagogy improves movement competence without compromising PA levels. I note that the discussion highlighted that all conditions were found to not meet recommended active time during PE. Would it be more accurate for the authors to acknowledge in the conclusion that all approaches failed to meet PA recommendations during PE instruction?

- A relatively minor point - the use of italics quite liberally particularly in the discussion tends to make reading more difficult. Probably, consider whether this approach is truly necessary.

Reviewers' comments:

Reviewer's Responses to Questions

**Comments to the Author**

1. Is the manuscript technically sound, and do the data support the conclusions?

Reviewer #1: Yes

2. Has the statistical analysis been performed appropriately and rigorously? 

Reviewer #1: Yes

3. Have the authors made all data underlying the findings in their manuscript fully available?

Reviewer #1: Yes

4. Is the manuscript presented in an intelligible fashion and written in standard English?

Reviewer #1: Yes

5. Review Comments to the Author

Reviewer #1: Overall, this was a very interesting and methodologically sound study. I only have a few comments for the authors to consider.

- On line 271, it is stated that the children’s gender was measured – I assume this was actually biological sex, not gender, that was captured? The manuscript goes on to use the term sex throughout, so I would suggest revision to be consistent.

- It would be helpful to readers if the specific accelerometer cut-points were identified/names (beyond a reference number) in the text so they don’t have to go searching in the reference list

- Please include a description of the missing data – whether it was missing in different proportions between groups or if it was related to any relevant variables.

- Can you provide an explanation or rationale for why baseline MVPA levels were not measured?

- The difference in the results between the unadjusted and adjusted model may suggest the possibility of selection bias and unbalanced groups (in unmeasured variables, i.e. MVPA baseline levels)

- Were there any samples size or power calculations conducted?

- The results section seems to repeat a lot of what is reported in the tables – I suggest editing to be more succinct, less verbose.

- I'm not sure I understand the purpose of this comparison between the current study and the PACES study for the outcome of % MVPA since they reported similar post-intervention data. Their results indicate that despite less time off task and more small-sided activities, %MVPA wasn't largely affected (beyond what you've reported in your results), so it may not be a mediating factor in the intervention effectiveness.

- The next paragraph describing the Weaver 2018 study: "significant improvements" suggests a pre-post change which you did not measure so this doesn't seem exactly directly comparable. Did Weaver et al 2018 find a significant difference between groups?

- In the section “Teaching practices in pedagogies underpinned by movement learning theories” there is a lot of repetition from the results – again, I would suggest revision to be more concise.

- It seems contradictory to be suggesting the use of strategies employed in both/either linear and nonlinear pedagogies considering that neither led to increases in MVPA

- It is not clear if lines 682-84 are referring to results from this study or if you are describing general patterns in the literature.

6. PLOS authors have the option to publish the peer review history of their article (what does this mean?). If published, this will include your full peer review and any attached files.

Reviewer #1: No

---

## [Author Response · Author response to Decision Letter 0]

20 May 2022

Response to reviewers

Reviewer 1

Comment 1

Reviewer #1: Overall, this was a very interesting and methodologically sound study. I only have a few comments for the authors to consider.

 Answer

We would like to thank Reviewer 1 for the constructive and positive feedback. Our responses to specific comments are outlined below.

Comment 2

- On line 271, it is stated that the children’s gender was measured – I assume this was actually biological sex, not gender, that was captured? The manuscript goes on to use the term sex throughout, so I would suggest revision to be consistent.

 Answer

This comment was addressed in line 296 by substituting “gender” with “sex”.

Comment 3

- It would be helpful to readers if the specific accelerometer cut-points were identified/names (beyond a reference number) in the text so they don’t have to go searching in the reference list

 Answer

We modified the following sentence in lines 311-312: 

“Lastly, age-appropriate cut-points by Crotti et al. (2020) were used to classify ENMO accelerations equal or higher than 189 mg into time spent in MVPA (68)”

Comment 4

- Please include a description of the missing data – whether it was missing in different proportions between groups or if it was related to any relevant variables.

 Answer

We added the following sentence in lines 396-400: 

“Parents did not report neighbourhood deprivation for 1 child in the control group, while ethnicity information was not provided for 2 children in the Linear pedagogy group and 2 children in the Nonlinear pedagogy group. Due to time constraints, we were not able to measure the BMI of 3 children from the Linear pedagogy group, 4 children from the Nonlinear pedagogy group and 9 from the Control group.”

Comment 5

- Can you provide an explanation or rationale for why baseline MVPA levels were not measured?

 Answer

As stated in lines 155, 179 and 279, this study was part of the process evaluation of the SAMPLE-PE randomised controlled trial, which is described in detail in the study protocol paper by Rudd et al. (2020). As a process evaluation, the focus of measurement was on the intervention delivery and implementation, rather than examining changes in outcomes (which will be reported elsewhere – manuscript in preparation). The process evaluation was designed to evaluate participants’ responsiveness in terms of MVPA levels in PE and PA promoting teaching practices among teachers within the SAMPLE-PE project. Therefore, baseline MVPA levels in PE were not measured. The results of this process evaluation paper are important to understand whether children’s MVPA and teaching practices during PE might have played a role in affecting the primary and secondary outcomes of the SAMPLE-PE cluster-RCT, including “motor competence”, “habitual physical activity” and “cognition” in children. These outcomes were measured at baseline and post intervention. 

Nevertheless, we agree with the reviewer that this point could be clarified. Therefore, we have added the following sentence in lines 158-161:

“Specifically, this study was designed to evaluate the implementation of the interventions and explore PA promoting teaching practices during PE lessons and participants’ responsiveness in terms of children’s MVPA levels, rather than to evaluate changes in these constructs from baseline to post-intervention.” 

Thus, increasing MVPA levels in PE was not a primary or secondary outcome in the SAMPLE-PE project. While the current study design is unable to determine the impact and effectiveness of Linear and NonLinear Pedagogy at increasing MVPA levels in PE and increasing PA promoting teaching practices, as reported in lines 144-145, no studies to date have examined and compared MVPA and teaching practices in Linear Pedagogy and Nonlinear Pedagogy approaches. Therefore, assessing the MVPA and teaching practices in PE would be a step forward to better understanding and identifying intervention strategies that motor competence focused approaches (such as Linear and Nonlinear pedagogy) could incorporate to better promote PA and optimise children’s MVPA engagement.

Comment 6

- The difference in the results between the unadjusted and adjusted model may suggest the possibility of selection bias and unbalanced groups (in unmeasured variables, i.e. MVPA baseline levels)

 Answer

Within this study the participating schools were randomly allocated to a group to prevent selection bias; therefore, any group differences in participant characteristics would purely be by chance. Furthermore, we suggest that the results of the fully adjusted and unadjusted models were different as the unadjusted models did not include important predictors of MVPA in children during PE. The fully adjusted models presented a significantly increased fit compared to the unadjusted models that is in line with the increased R squared and ICCs reported in the fully adjusted models compared to the unadjusted models. The fully adjusted models therefore better explain the variance in the physical activity levels of the children during PE. Therefore, the reasons why the variable “group” was not a significant predictor of MVPA within the fully adjusted models would be that other factors better predict MVPA levels in children compared to the “groups” variable (i.e., either Linear pedagogy, Nonlinear pedagogy or Control group). Examples of variables that better explained MVPA variance compared to “groups” would be “Lesson content”, “Lesson location” and “Lesson duration” while also the “teachers” variables explained a large chunk of variance within the models as reported in lines 651-655. To clarify that the fully adjusted models presented a better fit compared to the unadjusted model we modified the sentence in lines 359-361 as follows:

“During the modelling process, we decided to include variables that significantly increased the fit of the model and to exclude the nesting level of school class as it did not lead to an improved model fit or led to overfitted models.”

The group differences observed in the unadjusted analyses can be explained by the reverse of the above explanation – the lack of consideration for those factors such as lesson content, location or duration, as well as the group characteristics (which occurred by chance), likely account for significant differences, rather than the pedagogy per se. 

Comment 7

- Were there any samples size or power calculations conducted?

 Answer

The SAMPLE-PE project randomised controlled trial was powered as reported in the protocol paper by Rudd et al. (2020) to assess movement competence in 3 groups over 3 time points leading to the recruitment of 360 children in 12 schools.

A smaller sample was identified as being adequate for the purpose of the present process evaluation study by utilising a sub-sample of 50% of the children who provided consent to participate in the SAMPLE-PE project within 9 schools. However, we agree with the reviewer that further information about sample size and power should be provided. Therefore, we added the following text at lines 181-182:

“For feasibility and time constraint reasons and in line with sample size calculations reported below,”

And in lines 187- 202:

“Sample Size and Statistical Power

Sample size and power calculations for the SAMPLE-PE cluster-randomised controlled trial are reported elsewhere (52). For the purposes of this study, an a priory power calculation was undertaken using G*Power software to detect differences between 3 groups including a large effect size based on the review by Fairclough et al. (53), 90% power, alpha levels set at p < 0.05 and multiple covariates recommended a minimal sample size of 83 children. It was not possible to account for clustering factors (e.g. school) in the power calculation as the mixed model analysis reported in previous literature did not report ICCs associated with clustering factors. Previous studies that have assessed MVPA during PE included a sample size similar or higher than 83 children (e.g. up to 168 children) (54–59). Therefore, in line with the power calculation and the sample sizes observed in previous research, and after accounting for potential dropout, we aimed to recruit 50% of the research participants, amounting to 157 children, which was considered adequate for the purpose of this study (52). Due to the lack of previous research reporting effect sizes about SOFIT+ outcomes and feasibility factors such as time and resource constraints and school burden, we aimed to collect data about teaching practices in 3 lessons per class participating in the project.”

Comment 8

- The results section seems to repeat a lot of what is reported in the tables – I suggest editing to be more succinct, less verbose.

 Answer

We agree with the reviewer’s comment, therefore, we decided to remove the text on lines 490-504 in the original submission that described the percentages shown in Table 3. We deleted this text for the following reasons:

• The reader can easily observe information about the frequency of teaching practices (%) in Table 3.

• The text we deleted reported the teaching practices that were observed more frequently compared to others in each group. However, the text we deleted was a repetition of what we reported in the results section when describing the outcomes of the teaching practices statistical data analysis shown in Table 7.

We believe that the text between lines 546-570 should be kept in the manuscript as this text is key for the reader to understand and navigate Table 7, which might be perceived as data heavy by some readers, especially those who are not familiar with the statistical analysis we used (Negative Binomials).

Comment 9

- I'm not sure I understand the purpose of this comparison between the current study and the PACES study for the outcome of % MVPA since they reported similar post-intervention data. Their results indicate that despite less time off task and more small-sided activities, %MVPA wasn't largely affected (beyond what you've reported in your results), so it may not be a mediating factor in the intervention effectiveness.

 Answer

We understand the reviewer’s point that the post intervention percentage of MVPA reported in the PACES study were similar to the results reported in our study. However, it must be recognised that the PACES study intervention led to a significant increase in MVPA during PE that was higher than 10% in both males and females. Given that specific PA promoting teaching practices might have played a key role in the effectiveness of the intervention (i.e. significant increase of small sided games and verbal promotion of PA as well as decreased children off-task) and given that Linear and Nonlinear pedagogy interventions presented lower percentages of small-sided activities and verbal PA promotion as well as higher percentages of children being off task compared to the ones reported in the PACES study, we believe it is likely that improvements in these variables could lead in increased MVPA within future Linear and Nonlinear pedagogy interventions.

Comment 10

- The next paragraph describing the Weaver 2018 study: "significant improvements" suggests a pre-post change which you did not measure so this doesn't seem exactly directly comparable. Did Weaver et al 2018 find a significant difference between groups?

 Answer

We discussed the studies by Weaver et al. (i.e. PACES study and Weaver et al (2018)) as well as the SHARP study as they report MVPA during PE using accelerometers and assessed teaching practices using SOFIT+ or SOFIT, in line with the methods used in our study. We believe that the interpretation of our data compared to the results found in these studies can lead to interesting suggestions about how pedagogical interventions could be improved to promote MVPA in the future. However, we understand the reviewer’s concerns suggesting that the aforementioned studies utilised different design (pre-post) and aims (assessing improvement in PA) compared to our study. Therefore, to address this comment, we clarified the reasons for discussing the studies by Weaver and colleagues as well as the SHARP study within the “Increasing physical activity in physical education” discussion section. Furthermore, following the Reviewer’s comment we have condensed the subsequent paragraphs that gave an overview of the PACES, Weaver et al. and SHARP intervention findings and recommendations into a single paragraph in the revised manuscript, see lines 626-647:

“Although presenting different research design and aims compared to our study, lessons can be learned from some of the aforementioned primary school PE interventions that targeted the improvement of MVPA and PA promoting teaching practices and measured changes in these outcomes from baseline to post-intervention (17,57,58). Based on the findings from the Partnerships for Active Children in Elementary Schools (PACES) intervention study (17) we suggest that future Linear and Nonlinear pedagogy interventions aiming to improve children’s MVPA during PE could seek to increase Small Sided Activity as well as teacher Promotes PA time and reduce Children off task (i.e., time when one or more students are not engaged in the task proposed by the teacher). Furthermore, considering evidence from a follow-up to the PACES study by Weaver et al. (2018) (57) we advise that decreasing Knowledge time and increasing Motor Content time in future Linear and Nonlinear pedagogy interventions as well as decreasing Waiting Activity in future Linear pedagogy interventions could also be effective and feasible strategies to foster children’s MVPA in PE. Finally, the “SHARP” intervention (58) reported a significant increase in MVPA together with increased time in teaching practices such as Skill Practice and “in class PA promotion” within the intervention group compared to the control group. The increase in Skill Practice observed in the SHARP intervention could be associated with the SHARP principle concerning “high repetition of motor skills” that is also a key principle within the Linear pedagogical intervention delivered in this study suggesting that practicing movement skills can significantly contribute to MVPA in PE (58,85). Furthermore, the high percentages of verbal PA promotion within the SHARP (42.3%) intervention compared to that observed in this study (0-0.2%) confirms that future Linear and Nonlinear interventions could focus on improving verbal PA promotion during PE delivery as a strategy to improve children’s MVPA in PE (58).”

Comment 11

- In the section “Teaching practices in pedagogies underpinned by movement learning theories” there is a lot of repetition from the results – again, I would suggest revision to be more concise.

 Answer

We understand the point of the reviewer about trying to avoid repetition as much as possible. However, the 2 paragraphs focusing on Linear pedagogy (lines 690-706) and Nonlinear Pedagogy (lines 707-725) within the section “Teaching practices in pedagogies underpinned by movement learning theories” have been written with a specific structure and with the purpose to compare our findings with previous theories and literature:

• Each of the 2 paragraphs start with two sentences reporting that the teaching practices used in our interventions are in line with the principles of either teacher-centred (for Linear pedagogy) or learner-centred (for Linear pedagogy) principles. 

• The second part of each paragraph highlights that the teaching practices observed in either Linear or Nonlinear pedagogy interventions are in line with the pedagogical approaches being focused on movement competence promotion.

• The last part of the paragraphs focuses on what could be improved in terms of teaching practices to promote higher engagement in MVPA in future interventions. 

However, in order to make this structure clearer we modified two sentences in the text as reported below.

Line 690: “As expected from a teacher-centred pedagogical approach, the Linear pedagogy intervention”

Line 707: “As expected from a learner-centred pedagogical approach, the Nonlinear pedagogy”.

Comment 12

- It seems contradictory to be suggesting the use of strategies employed in both/either linear and nonlinear pedagogies considering that neither led to increases in MVPA

 Answer

We agree with the reviewer that suggesting Linear and Nonlinear pedagogy would guarantee to improve MVPA during PE would be a wrong conclusion based on our results. In fact, in our conclusion, we did not make a statement about Linear and Nonlinear pedagogy guaranteeing higher MVPA in PE but we reported the following:

“The majority of children’s MVPA levels within both intervention and control groups did not reach the recommended MVPA engagement of 50% within PE in line with previous literature. Furthermore, compared to current practice in PE, interventions based on Linear and Nonlinear pedagogy were not associated with increased children’s MVPA, but they included a higher incidence of MVPA promoting teaching practices (e.g., Motor content, Skill Practice, Discovery Practice). Nevertheless, the findings suggest that utilising Linear and Nonlinear pedagogies in PE could potentially improve movement competences in young children without compromising children’s PA levels compared to general practice. Given that PE deliverers were the main predictor of MVPA in PE in this study, future interventions should focus on improving the pedagogic knowledge and skills of PE deliverers about increasing children’s MVPA. This paper provides valuable information about how teaching practices within different pedagogical approaches affect PA in PE and proposes teaching practices that should be targeted to improve MVPA in PE. These findings can be used to help practitioners and researchers who are interested in designing future PE or coaching interventions based on Linear or Nonlinear pedagogies and/or maximizing MVPA engagement in PE.”

Comment 13

- It is not clear if lines 682-84 are referring to results from this study or if you are describing general patterns in the literature.

 Answer

We agree with the reviewer that the sentence is not clear. In the sentence we initially referred to previous literature and in the second part of the sentence we referred to the results of this study. Therefore, we modified the sentence in lines 700-703 as follows to clarify:

“However, within previous literature Game Play was found to be associated with the highest MVPA engagement in PE compared to other type of Lesson Contexts and within this study Game Play was observed less frequently in Linear intervention compared to the control group (14,69,79).”

Additional Editor Comments (if provided):

Comment 1

The study has potentially important contributions to the evidence related to physical education in relation to physical activity and motor competence of young children. There are areas in the manuscript that can be strengthened, which the authors might wish to consider:

 Answer

We would like to thank the editor for the constructive and positive feedback. Our responses to specific comments are outlined below. We also inform the editor that during the correction process we revised any inconsistency in the use of capital letters throughout the article.

Comment 2

- In the introduction, it was not clear why the researchers expected any differences in accrued PA between linear, non-linear, and usual practice conditions. While there linear and non-linear pedagogy were adequately explained, the underlying rationale for the research question was not made clear.

 Answer

We understand the editor’s concern. We believe that we indicated the reasons why pedagogies focused on movement competence development could be expected to foster high levels of PA engagement as outlined below:

Firstly, we raised the concern that current general practice in PE is often led by generalist teachers who might not have the skills to foster specific learning outcomes as well as PA engagement in PE, as reported in Lines 75-77:

“However, there are concerns that primary PE deliverers (which often include generalist classroom teachers) do not have the required level of pedagogic content knowledge to support learning and foster student’s PA (26).”

Subsequently, we suggested that there is lack of research about how different pedagogical approaches affect PA in PE in Lines 77-80:

“Nevertheless, few studies have examined the association between different pedagogical approaches in PE and student MVPA levels. Thus, to maximise PA opportunities during PE, examining the extent to which teaching practices support students’ MVPA under different pedagogical conditions is warranted.”

Then we argued that for the development of movement skills it is necessary for children to engage in PA, so potentially pedagogies that maximize movement competence learning would be maximizing PA engagement as well, as stated in Lines 85-92:

“However, movement skills do not develop by maturation alone, children need to be physically active within favourable conditions for movement skills to emerge and progress, such as through structured teaching and learning activities (29). The more a child moves the greater the opportunity to develop and acquire competence in movement skills (30,31), which should lead to enhanced engagement in PA (27,30,31). Therefore, from a PE perspective, pedagogical approaches aimed at fostering movement competence should also seek to maximise opportunities for students to be physically active.”

Next, we talked about pedagogical approaches and how they differ in terms of their approaches to movement development as we clarified before that movement development and PA are strongly intertwined. Lines 95-97:

“Linear and Nonlinear pedagogy are two pedagogical approaches underpinned by different theories of motor learning that can guide the design of PE lessons aiming to foster the development of movement competence.”

Lastly, in the final part of the discussion, we again remind the reader that movement development emerges through PA engagement to remind the reader why pedagogical approaches focused on movement competence could foster high levels of PA in PE and that we would compare these approaches to current PE practice. 

However, we understand the reviewer’s point about clarifying the rationale behind this study therefore modified lines 145-147 as follows:

“Furthermore, no study evaluated whether Linear and Nonlinear pedagogy would be associated with higher children’s MVPA and PA promoting teaching practices compared to current PE.”

Comment 3

- Is there a sense of competition in PE for promoting motor competence vs. PA accrual? This was taken up in the discussion, but I believe this needs to be taken up earlier in the introduction.

 Answer

We believe we talked about the relation between PA accrual and movement competence development in different parts of the introduction section: For example, in lines 61-63 we reported:

“While it is important to acknowledge that the focus on MVPA should not come at the expense of other important and meaningful PE learning outcomes (18,19)”

 And later in lines 81-83 we reported that one of the important learning outcomes in PE is movement competence development:

“An important feature of meaningful PE experiences and a key objective for PE curricula in young children (5-to-7-years-old) is the development of foundational movement skills needed for a lifetime of diverse PA opportunities (1,5,18).”

Therefore, we believe that within the introduction section we clarified to the reader that PA accrual should not come to the expenses of motor competence learning. Conversely, we explained that “children need to be physically active within favourable conditions for movement skills to emerge and progress, such as through structured teaching and learning activities (29).” In lines 86-88.

Comment 4

- There was discussion of related studies such as SHARP, PACES - however, there was no critical discussion of how the current findings contribute/expand the relevant evidence base. I believe there is room for the authors to develop a more critical and insightful discussion to place their findings in the context of the wider evidence base.

 Answer

Firstly, it is important to highlight that this is the first study to examine MVPA levels in PE under movement competence pedagogical approaches, and the first to examine PA promoting teaching practices within Linear and Nonlinear pedagogies. Therefore, the current findings make a novel and original contribution to the literature. 

In addition, we believe that we offered extensive explanation in the discussion about how our findings compared to previous literature and could extend previous research findings as explained below:

Within the section, “Increasing physical activity in physical education”, in lines 593-615, we explained how our MVPA findings can be compared to findings from extensive literature about MVPA in PE.

Additionally, in the section titled “Factors associated with children’s physical activity in physical education” within lines 651-683 we compared and critically discussed our findings in comparison with several findings from previous literature.

Within the section “Teaching practices in pedagogies underpinned by movement learning theories”, we critically discussed how our findings could inform future PE interventions, PA promotion in PE and teaching practices in PE.

Lastly, within the “Strengths and limitations” section in lines 731-739 we further clarified how our study extended findings from previous literature.

However, following the reviewer’s suggestion, to further improve the discussion section we decided to include the following paragraph in the section “Increasing physical activity in physical education” (see lines 615-625) suggesting how our findings would expand relevant evidence about PE interventions focusing on improving both motor competence and PA during PE in children: 

“Furthermore, of the studies assessing PA in PE, our study was the first reporting the pedagogical basis guiding the delivery of movement learning activities. As an example, the “Move it Groove it” and “PLUNGE” interventions reported both PA and movement skills development as aims of their PE interventions (54,84). However, despite describing strategies to improve MVPA in PE, neither of these two studies clarified the pedagogical basis guiding the delivery of movement learning activities (54,84). Therefore, we suggest that future research should further investigate how different pedagogies and PA promotion strategies might affect children’s PA during PE. Furthermore, we recommend that clear descriptions of pedagogies and PA promotion strategies should be reported in future PE interventions studies as this could help both practitioners and researchers understanding how to achieve and/or prioritise specific PE outcomes (e.g. children’s motor competence development or high MVPA engagement).”

Comment 5

- I wonder about the conclusion where the authors say that linear and non-linear pedagogy improves movement competence without compromising PA levels. I note that the discussion highlighted that all conditions were found to not meet recommended active time during PE. Would it be more accurate for the authors to acknowledge in the conclusion that all approaches failed to meet PA recommendations during PE instruction?

 Answer

We agree. Therefore, we modified lines 765-772 as follows:

“The majority of children’s MVPA levels within both intervention and control groups did not reach the recommended MVPA engagement of 50% within PE in line with previous literature. Furthermore, compared to current practice in PE, interventions based on Linear and Nonlinear pedagogy were not associated with increased children’s MVPA, but they included a higher incidence of MVPA promoting teaching practices (e.g., Motor content, Skill Practice, Discovery Practice). Nevertheless, the findings suggest that utilising Linear and Nonlinear pedagogies in PE could potentially improve movement competences in young children without compromising children’s PA levels compared to current practice.”

Comment 6

- A relatively minor point - the use of italics quite liberally particularly in the discussion tends to make reading more difficult. Probably, consider whether this approach is truly necessary.

 Answer

Within the initial draft of the article, we did not use “Italics” to highlight teaching practices and as a consequence we found it difficult to distinguish when the article was talking about SOFIT+ teaching practices versus when the article was talking about other general PE related aspects. Therefore, we suggest that using the “Italics” font will help readers navigate the article more easily.

---

## [Decision Letter · Decision Letter 1]

5 Jul 2022

PONE-D-21-38034R1Physical activity promoting teaching practices and children’s physical activity within physical education lessons underpinned by motor learning theory (SAMPLE-PE)PLOS ONE

Dear Dr. Foweather,

Thank you for submitting your manuscript to PLOS ONE. After careful consideration, we feel that it has merit but does not fully meet PLOS ONE’s publication criteria as it currently stands. Therefore, we invite you to submit a revised version of the manuscript that addresses the points raised during the review process.

The revision generally responded positively to the comments raised in the previous review. However, the reviewer noted that there are still a couple of minor points that can be addressed prior to publication. The authors might consider these minor issues.

We look forward to receiving your revised manuscript.

Kind regards,

Catherine M. Capio

Academic Editor

PLOS ONE

Journal Requirements:

Reviewers' comments:

Reviewer's Responses to Questions

**Comments to the Author**

1. If the authors have adequately addressed your comments raised in a previous round of review and you feel that this manuscript is now acceptable for publication, you may indicate that here to bypass the “Comments to the Author” section, enter your conflict of interest statement in the “Confidential to Editor” section, and submit your "Accept" recommendation.

Reviewer #1: (No Response)

2. Is the manuscript technically sound, and do the data support the conclusions?

Reviewer #1: Yes

3. Has the statistical analysis been performed appropriately and rigorously? 

Reviewer #1: Yes

4. Have the authors made all data underlying the findings in their manuscript fully available?

Reviewer #1: Yes

5. Is the manuscript presented in an intelligible fashion and written in standard English?

Reviewer #1: Yes

6. Review Comments to the Author

Reviewer #1: Thank you for addressing all of my previous concerns and comments. I believe the manuscript is much improved and nearly ready for publication. I have a two remaining issues, I'd like to see resolved prior to endorsing publication.

- Regarding your response to comment 5, I think part of the confusion stems from the use of the word, ‘responsiveness’ which generally implies a change. By measuring PA levels once, you really can’t measure responsiveness since we don’t know if these PA levels are responding to the intervention or if they are similar to baseline levels

- My confusion regarding your statement about your results suggesting that linear pedagogy could help achieve increased MVPA in PE was specific to your sentence on line 699 -700. Perhaps just make it clearer that although it was not demonstrated in this study, increases in time spent in motor content and teacher PA engagement may over time lead to increases in MVPA. It remains confusing why the text appears to be concluding that either pedagogy may lead to increased MVPA when your results do not support this.

7. PLOS authors have the option to publish the peer review history of their article (what does this mean?). If published, this will include your full peer review and any attached files.

Reviewer #1: No

---

## [Author Response · Author response to Decision Letter 1]

7 Jul 2022

Reviewer 1

Comment 1

Thank you for addressing all of my previous concerns and comments. I believe the manuscript is much improved and nearly ready for publication. I have a two remaining issues, I'd like to see resolved prior to endorsing publication.

 Answer

We would like to thank Reviewer 1 for the constructive and positive feedback. Our responses to specific comments are outlined below.

Comment 2

- Regarding your response to comment 5, I think part of the confusion stems from the use of the word, ‘responsiveness’ which generally implies a change. By measuring PA levels once, you really can’t measure responsiveness since we don’t know if these PA levels are responding to the intervention or if they are similar to baseline levels

 Answer

To clarify the meaning of the word “responsiveness” we modified the sentence in lines 158-162 by adding the definition of responsiveness in brackets together with a relevant citation:

“Specifically, this study was designed to evaluate the implementation of the interventions and explore PA promoting teaching practices during PE lessons and participants’ responsiveness (that concerns the measurement of how far participants respond to, or are engaged by, an intervention (53)) in terms of children’s MVPA engagement, rather than to evaluate changes in these constructs from baseline to post-intervention.”

As suggested in the sentence in lines 158-162, the assessment of responsiveness in our paper falls under the definition: how far participants are engaged by an intervention.

The article we cited was:

Carroll C, Patterson M, Wood S, Booth A, Rick J, Balain S. A conceptual framework for implementation fidelity. Implement Sci. 2007

Comment 3

- My confusion regarding your statement about your results suggesting that linear pedagogy could help achieve increased MVPA in PE was specific to your sentence on line 699 -700. Perhaps just make it clearer that although it was not demonstrated in this study, increases in time spent in motor content and teacher PA engagement may over time lead to increases in MVPA. It remains confusing why the text appears to be concluding that either pedagogy may lead to increased MVPA when your results do not support this.

 Answer

To address any confusion in the interpretation of our findings we deleted the sentences that were in lines 699-700 and in lines 715-717 in our previous submission and we added the following sentence at the end of the “Teaching practices in pedagogies underpinned by movement learning theories” section in lines 726-729:

“Nevertheless, taking all the above findings together, the results suggest that Linear and Nonlinear pedagogical interventions both improve time allocated to movement competence practice but would need to adopt more PA promoting teaching practices to increase children’s MVPA in PE (70,80,81).”

We also deleted the words “Skill Practice and” in line 694 as “Skill Practice” was mentioned already in the previous sentence.

---

## [Editor Report · Decision Letter 2]

19 Jul 2022

Physical activity promoting teaching practices and children’s physical activity within physical education lessons underpinned by motor learning theory (SAMPLE-PE)

PONE-D-21-38034R2

Dear Dr. Foweather,

We’re pleased to inform you that your manuscript has been judged scientifically suitable for publication and will be formally accepted for publication once it meets all outstanding technical requirements.

Kind regards,

Catherine M. Capio

Academic Editor

PLOS ONE
---

## [Editor Report · Acceptance letter]

21 Jul 2022

PONE-D-21-38034R2 

Physical activity promoting teaching practices and children’s physical activity within physical education lessons underpinned by motor learning theory (SAMPLE-PE) 

Dear Dr. Foweather:

I'm pleased to inform you that your manuscript has been deemed suitable for publication in PLOS ONE. Congratulations! Your manuscript is now with our production department. 

Kind regards, 

on behalf of

Dr. Catherine M. Capio 

Academic Editor

PLOS ONE